# T_h_17, T_h_22, and Myeloid-Derived Suppressor Cell Population Dynamics and Response to IL-6 in 4T1 Mammary Carcinoma

**DOI:** 10.3390/ijms231810299

**Published:** 2022-09-07

**Authors:** Viva J. Rasé, Reid Hayward, James M. Haughian, Nicholas A. Pullen

**Affiliations:** 1School of Biological Sciences, University of Northern Colorado, Greeley, CO 80639, USA; 2Department of Pathology, University of Utah School of Medicine, Salt Lake City, UT 84112, USA; 3School of Sport and Exercise Science, University of Northern Colorado, Greeley, CO 80639, USA

**Keywords:** type 3 immunity, MDSC, IL-22, cancer

## Abstract

Immunotherapies relying on type 1 immunity have shown robust clinical responses in some cancers yet remain relatively ineffective in solid breast tumors. Polarization toward type 2 immunity and expansion of myeloid-derived suppressor cells (MDSC) confer resistance to therapy, though it remains unclear whether polarization toward type 3 immunity occurs or has a similar effect. Therefore, we investigated the involvement of type 3 T_h_17 and T_h_22 cells and their association with expanding MDSC populations in the 4T1 mouse mammary carcinoma model. T_h_17 and T_h_22 were detected in the earliest measurable mass at d 14 and remained present until the final sampling on d 28. In peripheral organs, T_h_17 populations were significantly higher than the non-tumor bearing control and peaked early at d 7, before a palpable tumor had formed. Peripheral T_h_22 proportions were also significantly increased, though at later times when tumors were established. To further address the mechanism underlying type 3 immune cell and MDSC recruitment, we used CRISPR-Cas9 to knock out 4T1 tumor production of interleukin-6 (4T1-IL-6-KO), which functions in myelopoiesis, MDSC recruitment, and T_h_ maturation. While 4T1-IL-6-KO tumor growth was similar to the control, the reduced IL-6 significantly expanded the total CD4^+^ T_h_ population and T_h_17 in tumors, while T_h_22 and MDSC were reduced in all tissues; this suggests that clinical IL-6 depletion combined with immunotherapy could improve outcomes. In sum, 4T1 mammary carcinomas secrete IL-6 and other factors, to polarize and reshape T_h_ populations and expand distinct T_h_17 and T_h_22 populations, which may facilitate tumor growth and confer immunotherapy resistance.

## 1. Introduction

Immune checkpoint therapies targeting CTLA-4 and PD-1/PD-L1 have shown clinical success in hematological tumors and solid tumors with a high neoantigen burden, leading to tumor regression and considerable gains in patient survival [1,2,3,4,5]. Unfortunately, not all patients benefit equally from checkpoint inhibition as non-responders or poor responders, adverse immune events, and tumor hyper-progression have also been reported [6,7,8,9,10]. The reports of hyper-progression suggest, paradoxically, that releasing checkpoint controls on the immune system can aid in tumor progression in some immunological circumstances. These undesirable outcomes clearly indicate a need to understand better the cancer patient’s immunological status, to ensure that checkpoint and other immunotherapies are delivered into a well-defined and optimally effective immune context. 

Tumor responses to checkpoint immunotherapy are undoubtedly influenced by the phenotype and quantity of localized tumor infiltrating lymphocytes (TILs) [11], and the type 1 T_h_1 cell phenotype is generally accepted as the most effective coordinator against tumor cells in this locale, as it works in conjunction with most current immunotherapy treatments [12,13]. However, it is now clear that tumors signal beyond their physical boundaries and release systemic factors that mitigate the type 1 T cell response and foster immune responses beneficial to the tumor. For example, tumors may secrete G-CSF, M-CSF, or IL-6 cytokines to dysregulate myelopoiesis and promote the production of immune inhibitory cells, such as MDSC, that impair CD8^+^ cytotoxic T cell activation and more broadly inhibit type 1 effector T_h_ cell function [14]. 

Another mode of systemic immune evasion is tumor-dependent immune polarization that attenuates the type 1 response and intensifies the potentially tumor-promoting type 2 or type 3 T_h_ response [15,16,17,18]. Evidence from preclinical models such as MMTV-PyMT mammary and other cancer types indicates that tumors can skew T cell populations toward a helminth-targeting type 2 (T_h_2 cell-mediated) response [19,20,21]. Similarly, T cell polarization toward a T_h_17- or T_h_22-cell-mediated type 3 response that would normally combat microbial infection has been observed in mouse models and human cases of cervical, ovarian, prostate, and gastric cancers [16,22,23,24,25,26,27]. Specifically in mammary carcinoma, IL-22, a cytokine produced in large quantities by T_h_22, has been shown to exacerbate tumor growth [28]. Intriguingly, polarization toward type 3 immunity in other conditions, including *H. pylori* infection and autoimmune disease (lupus erythematosus, encephalomyelitis, arthritis), arises concurrently with elevated MDSC [29,30,31], which is a well-described feature of many breast malignancies [32]. To our knowledge, no studies of mammary carcinoma have addressed the association between type 3 immunity and MDSC. 

Mammary carcinomas stand to benefit from type 3 immune polarization. The polarization toward type-3-immune T_h_17 requires the presence of TGFβ-1, IL-6, and IL-23; alternatively, T_h_22 requires the combination of IL-6, IL-23, and IL-1β [33]. The expansion of class-defining T_h_17 or T_h_22 helper cells could locally and systemically increase concentrations of the cytokines IL-17A and IL-22, respectively [34]. The chronic presence of these type 3 cytokines has been linked to increased tumor growth, enhanced tumor cell survival, and metastatic potential [25,35,36]. Similarly, knocking out the IL-17 receptor in the 4T1 model heightened apoptosis and slowed tumor cell proliferation [37]. Mechanistically, IL-6 is implicated in both MDSC recruitment and T_h_17/ T_h_22 polarization [33,38], yet no studies have explored whether chronic overexpression of IL-6 from a developing tumor may, in fact, elicit both MDSC and type 3 T_h_ expansion in the context of mammary carcinoma. 

The present study examines the response of type 3 T helper populations T_h_17 and T_h_22, along with transitional T_h_1/17 [39], in the 4T1 model of mouse mammary carcinoma. The 4T1 model shares similarities with metastatic triple-negative breast cancers that are resistant to checkpoint immunotherapy [40,41]. Moreover, 4T1 tumors elicit a pronounced myeloid reaction in the form of MDSC [42], which are regulated by IL-6 [43]. Thus, this system enabled the characterization of IL-6 and its role in co-regulating T helper and MDSC immunity. Our findings suggest there is a robust, IL-6-dependent polarization toward type 3 immunity, which accompanies MDSC expansion in this mammary carcinoma model, and that this altered T_h_ cell landscape should be considered to improve immune and other oncotherapies. 

## 2. Results 

### 2.1. T_h_ Populations in Response to 4T1-WT Tumors 

**Tumors and spleen.** Following injection of 10^4^ 4T1 carcinoma cells into the mammary glands of BALB/cJ mice, small palpable tumors formed at the injection site by d 14, and by ~d 28 tumors approached the maximum humane dimension allowable under our IACUC protocol (Figure 1A,B). During this course of tumor development, we also measured spleen mass, as reports have indicated splenomegaly is associated with late-stage, established 4T1 tumors [44]. Unexpectedly, at d 7, before there was a palpable tumor, we observed a significant reduction in spleen mass relative to the controls (Figure 1C). As the tumors became larger and more established (d 21 and 28), the tumor-bearing animals showed a pronounced increase in spleen size, consistent with what has been reported elsewhere. To our knowledge, this is the first report of an early reduction in spleen size in the 4T1 tumor model, which will require further studies to determine the underlying cause. 

**Total CD4^+^ T_h_ cell populations**. Using the gating strategy outlined in Figure 2, we initially assessed T_h_-immune cell populations in the spleen, blood, bone marrow, and tumor (when possible) at various stages of wild-type 4T1 (4T1-WT) mammary tumor development (Figure 3). The percentage of total CD4^+^ T_h_ cells remained relatively unchanged in the tumor, blood, and other peripheral tissues, with modest increases detected in the spleen and bone marrow at d 14 and 21, respectively (Figure 3A,B).

**Type 3 T_h_ cell populations**. While the total T_h_ cell population showed only modest changes over time, we observed dynamic changes in the T_h_ cell subtypes for type 1 and type 3 immunity. Both type 3 T_h_17 and T_h_22 cells were present in the tumor at all time points, where there was sufficient tissue to assess, with T_h_17 significantly declining over time, whereas T_h_22 increased in the tumor tissue (Figure 3D,F). Interestingly, in the spleen, the T_h_17 population increased significantly by d 7 post-4T1 injection (before a palpable tumor was detectable) and declined thereafter, such that the percentage of T_h_17 remaining by 28 d was significantly lower than in the control animals. This same trend in early T_h_17 recruitment and later decline was also observed in the blood and bone marrow (Figure 3C). Notably, the T_h_22 populations moved independently of T_h_17 and were not expanded at early time points in peripheral organs. T_h_22 showed a delayed increase and was only elevated at later time points, a pattern most prominent in the blood (Figure 3E). Intrigued by this inverse pattern of T_h_17 and T_h_22 recruitment, we computed the Pearson correlation between T_h_22 and T_h_17 recruitment in the various tissues over time. There was no correlation between T_h_22 and T_h_17 in the blood or bone marrow; however, there was a significant negative correlation (r^2^ = 0.14, *p* < 0.05) in the spleen and a significant positive correlation associated with their coexistence in tumors (r^2^ = 0.25, *p* < 0.05; Figure 3A–D). 

**T_h_1 polarization**. We also assessed T_h_1 recruitment, as type 1 immunity is centrally involved in cancer immunotherapies. In the blood (Figure 4G, center panel), T_h_1 was significantly expanded across all times post-injection, consistent with a mobilization of this T_h_ subtype in response to a neoplastic threat; however, this peaked early at d 7 and declined thereafter. A similar pattern was observed in bone marrow, except that the peak occurred at d 14 (Figure 4G, right panel). T_h_1 was likely mobilizing from the spleen, as proportions there were below controls at d 7, but then later recovered (Figure 4G, left panel). Interestingly, T_h_1 is present in the tumor, though the proportions did not significantly increase along with the peripheral rise in T_h_1, suggesting unknown tumor-specific mechanisms that were preventing infiltration of these cells into the tumor microenvironment (Figure 4H). 

To look for corresponding changes in the T helper populations, we correlated the proportions of T_h_1, T_h_17, and T_h_22 and found significant negative correlations between T_h_1 and T_h_17 in the spleen and tumor (Figure 4E,H) but a significant positive correlation in the bone marrow (Figure 4G). When assessing T_h_1 vs. T_h_22, there was a significant negative correlation between these two cell types in the tumor (Figure 4L). These findings suggest that in the spleen or tumor there is potential for polarization away from type 1 toward a type 3 T_h_ phenotype, particularly to T_h_22. In addition, it may be that the spleen and tumor are conducive to initial T_h_ activation and polarization (or repolarization), which may explain the relationships in these tissues, but not in blood (Figure 4F,J).

**Transitional type 1/3 T_h_1/17 populations**. Since we were interested in T_h_ skewing from type 1 (T_h_1) to type 3 (T_h_17 and T_h_22) or vice versa, we also quantified the T_h_1/17 transitional population, which represents T_h_17 cells transiting to and from an effective, anti-tumor type 1 T_h_1 phenotype [45]. In the spleen, T_h_1/17 were significantly elevated, but were relatively unchanged in blood and bone marrow throughout the time course (Figure 3I). T_h_1/17 were detectable in the tumor at each time point, with initially higher levels declining by d 28 (Figure 3J). Given the differentiation plasticity that exists among T_h_1, T_h_1/17, and T_h_17, it was interesting that the pattern of tumor-resident T_h_1/17 and T_h_17 was similar across the various time points. This relationship in the tumor was supported by a strongly positive Pearson correlation (r^2^ = 0.62, **** *p* < 0.00001, Figure 4P); however, a similar relationship was not observed in peripheral organs (Figure 4, M–O). Consistent with differentiation away from the type 1 T_h_1 cells toward the type 3 T_h_17 phenotype, we saw robust negative correlations between these populations in the spleen (r^2^ = 0.19, ** *p* < 0.001), blood (r^2^ = 0.36, *** *p* < 0.0001) and bone marrow (r^2^ = 0.71, **** *p* < 0.00001; Figure 4Q–S) and a similar non-significant trend in the tumor (Figure 4T). 

**MDSC expansion and relationship to type 3 T_h_17 and T_h_22.** Type 3 immune responses have been associated with MDSC in other diseases including lupus erythematosus, autoimmune encephalomyelitis, autoimmune arthritis, *H. pylori* infections, and colon cancer [29,30,31], so we wanted to determine if a similar association might exist in the 4T1 mammary carcinoma model. As expected in tumor-bearing animals [46], MDSC were expanded in the peripheral organs with the M-MDSC and PMN-MDSC subtypes exhibiting similar patterns (Figure 5A,B). In the spleen, MDSC increased progressively over time, while blood MDSC remained low until it increased precipitously during late-stage tumor development (d 28). In the bone marrow, MDSC peaked early at d 7 and then remained significantly elevated above the baseline across all remaining time points. Both M-MDSC and PMN-MDSC were detected in the tumors, but only the PMN-MDSC significantly increased over time (Figure 5A,B).

Again, we used Pearson correlations to compare the patterns of various T_h_ cell types described in Figure 3 across the time course of MDSC proportions (Figure 6). We observed a very strong positive association between M-MDSC and PMN-MDSC and T_h_17 in the bone marrow (Figure 6C,G) but no relationship between T_h_22 and MDSC in this tissue (Figure 6K,O). No significant correlations were observed in other tissues, except in blood, where there was an unexpected negative correlation between T_h_17 and either PMN- or M-MDSC (r^2^ = 0.21, * *p* < 0.05, and r^2^ = 0.42, * *p* < 0.05, respectively; Figure 6B,F).

### 2.2. Tumor IL-6 Knock Out Effects on Type 3 T_h_ Cell Recruitment 

**Tumor IL-6 stimulation of type 3 T_h_17 and T_h_22 cells.** Our time-course data suggested that MDSC and type 3 T_h_ cells exist at elevated proportions in systemic immunological tissues of tumor-bearing animals, specifically, both MDSC and T_h_17 were elevated in blood and bone marrow. Furthermore, there was some indication from the bone marrow that MDSC presence was influencing T_h_17 polarization. Collectively, however, there was little direct evidence from the correlative time course studies that MDSC (Figure 5) and the factors driving its recruitment are also responsible for eliciting the observed changes in type 3 T_h_ populations (Figure 3). Thus, we sought a more direct experimental manipulation to explore potentially linked regulation and the possible interdependence of these cell types. To this end, we noted that others have shown both MDSC and type 3 T_h_ populations are responsive to circulating IL-6 [33,43], thus we used CRISPR-Cas9 to knock out the IL-6 gene from the 4T1 cell line (4T1-IL6-KO; Figure 7A,B).

Unlike other studies, in which IL-6 is knocked out in the entire animal [47], this approach narrows the IL-6 deficiency to the 4T1 tumor cells responsible for eliciting the MDSC and type 3 T_h_ responses. Indeed, there remained native stromal and immune cells that continued to express IL-6 in whole tumors (Figure 7C). We did not observe a significant difference in tumor size or mass between the 4T1-WT and 4T1-IL6-KO lines (Figure 8A,B), but, interestingly, spleen mass was significantly higher in 4T1-IL6-KO compared to 4T1-WT animals (Figure 8C).

We focused on type 3 cells accumulated at the later d 28 time point, when tumors were well-established. The level of type 3 T_h_ infiltrating the tumors was intriguing, as 4T1-IL6-KO tumors had expanded T_h_17 (Figure 9B), while T_h_22 was significantly reduced (Figure 9D). These data suggest that both T_h_17 and T_h_22 are sensitive to the IL-6 secreted locally from the tumor cells, but in an opposing fashion. This observation may be in line with the capacity of the tumor microenvironment and its cytokine milieu to re-polarize both the type and subset of T_h_ cells [16]. In the peripheral tissues, the lack of tumor IL-6 at d 28 reduced T_h_17 (Figure 9A), and T_h_22 was also lowered (Figure 9C). This suggests that systemically secreted IL-6 from the tumor is necessary to sustain the type 3 T_h_ immune response outside the tumor.

**Tumor IL-6 effects on total T_h_, T_h_1, T_h_1/17, and MDSC.** Similar to our earlier time course study, we examined other T_h_ and MDSC populations in the 4T1-IL6-KO model at d 28. It is worth noting that the response of these cell populations could be viewed as indicators of how clinically depleting tumor IL-6 with targeted agents such as tocilizumab could influence the efficacy of checkpoint immunotherapies. For example, the proportion of total T_h_ cells was significantly increased in the blood and tumors of 4T1-IL6-KO tumor-bearing mice (Figure 10), an effect that could prove beneficial if translated to the oncology clinic. Levels of T_h_1/17 and T_h_1 were variable across the tissues and treatment groups (Figure 11). We saw T_h_1/17 significantly expanded in the blood and tumor of the 4T1-IL6-KO animals (Figure 11A), while T_h_1 was significantly increased in the spleen but significantly decreased in the blood of 4T1-IL6-KO tumor-bearing mice (Figure 11B).

The 4T1-specific knockout of IL-6 led to a significant reduction in both M-MDSC and PMN-MDSC in all peripheral organs (Figure 12A,C), and tumor=-infiltrating M-MDSC was also reduced in the 4T1-IL6-KO tumors (Figure 12B). This further supports the idea that tumor-derived IL-6 promotes MDSC expansion [43] and that blocking systemic IL-6 signaling could prove beneficial to cancer patients, so it should be further explored. Relating this decline in MDSC back to the concurrent changes in type 3 T_h_ populations, there is again some evidence that MDSC may be similarly regulated and perhaps responsible for changes in the T_h_22 populations, as these patterns very closely mirror each other in the IL-6 KO condition. However, any associated changes in T_h_17 did not parallel the reduced MDSC population, suggesting independent regulation of this T_h_ subtype.

## 3. Discussion

Our data support a model (graphical abstract) in which mammary tumors repurpose secreted IL-6 and other cytokines to act locally and systemically in reshaping normal T_h_ and myeloid immunity. This influence occurs quickly, as we found that type 3 T_h_17 and T_h_22 cell populations, which are more commonly associated with extracellular microbial infection [48], were recruited before formation of a visible or palpable mass, within 7 d of injecting 10^4^ cells (Figure 3). We have yet to determine if these early responses are IL-6-dependent, as our 4T1-IL6-KO studies focused on late-stage tumor development. As tumors become more established, the type 3 immune response appears to switch from a predominantly T_h_17 response to a T_h_22 response, and this switch is again driven in part by IL-6 (Figure 9) and likely other cytokines. The distinct timing and pattern of T_h_17 vs. T_h_22 recruitment reinforces that these are indeed distinct T_h_ lineages. Our data support previous work showing that tumor IL-6 impacts myelopoiesis and MDSC [43], but importantly it also positions tumor-derived IL-6 as a negative regulator of the total CD4^+^ T_h_ cell population, an effect that is particularly pronounced locally in the tumor (Figure 10). Additional grafting studies are required to determine whether there are direct interactions between MDSC and type 3 T_h_ immune cells. 

**Implications of type 3 polarization in breast cancer**. Our preclinical findings suggest that mammary tumors may elicit a type 3 T_h_ immune reaction, with the potential to diminish initial immune surveillance and ultimately reduce the efficacy of immunotherapies, including checkpoint inhibitors. While the 4T1 model is known to resist checkpoint therapies [49], further experiments are needed to determine the role played by polarization to pro-tumor type 2 (T_h_2) [50] or type 3 (T_h_17 or T_h_22) cells in this resistance. As others have suggested [16,51], a type 3 T_h_ immune response is expected to support tumor growth, while distracting away from an effective type 1 response. In the context of checkpoint-inhibitor treatment, T cell clonal expansion followed by polarization is possible; however, it remains to be seen which antigens these T_h_ cells recognize. It may be that they do not recognize tumor-specific neoantigen displayed on MHC and are merely bystanders, but, again, the pro-tumor attributes ascribed to these populations suggests otherwise. For example, T_h_22 cells may facilitate metastasis in high-grade breast cancer via IL-22 [52,53]. Moreover, IL-22 is known to promote wound healing in the skin and gastrointestinal tract and has been shown to promote mouse mammary cancer growth [28], all suggesting that this cytokine may support a breast tumor microenvironment and provide aid for the growth of malignant cells [35,54]. The 4T1 tumor model is representative of metastatic, high-grade breast cancer [40,41], which shows robust T_h_22 recruitment as time progresses (Figure 3). While tumor growth was similar with reduced T_h_22 in the 4T1-IL6-KO, the potential effects of T_h_22 and, by extension, IL-22 on metastatic burden warrant further study [35]. In addition to IL-22, we observed IL-17^+^ T_h_17 cells. We found that T_h_17 cells appeared to be present in high proportions in the bone marrow and tumor. This illustrates the necessity to further investigate the clinical consequences of recruited type 3 immunity in breast cancer. 

**MDSC**. The pro-tumor activities of these immature myeloid cells have been well-established [32,43,55,56]. In our time course study, early MDSC expansion in the bone marrow was followed by progressive increases in the spleen, suggesting gradual splenic takeover of MDSC maintenance (Figure 5A,B). The initial expansion of MDSC in the bone marrow is logical as this is the presumed tissue of origin, but it is remarkable that the MDSC increase was detectable very early (first sampling at d 7 post-engraftment), before a tumor can be visualized. A similar MDSC increase was seen in the spleen beginning on day 7, and it is interesting to note that the spleen mass is significantly lower than in non-tumor-bearing controls at this point, which might suggest substantial immune remodeling within the spleen before the tumor mass is established and splenomegaly ensues (Figure 5). A better understanding of these dynamic MDSC changes in human cancers could someday help inform clinicians about the stage of disease progression and the immune context of proposed cancer treatments [32]. 

It remains unclear, in the context of cancer, whether there are direct functional interactions between MDSC and type 3 immune cells. We saw varying correlations among these cell populations depending on cell and tissue type (Figure 7B,C,F,G). Speculating based on timing, our data may suggest that early MDSC in the bone marrow, which gas not acquired conventional suppressive capacity [57,58], may still be capable of locally influencing polarization toward type 3 T_h_17 responses, before leaving to mature in the spleen. While more studies are needed to confirm this relationship, this would be an additional novel mechanism by which MDSC could deplete the type 1 immune response and facilitate tumor progression. This speculative mechanism aligns with research in rheumatoid arthritis where MDSC promote T_h_17 responses, but are unable to suppress effector T_h_ to halt an autoimmune reaction [31]. Previously our lab has verified that both M-MDSC (CD11b^+^Ly-6G^−^Ly-6C^hi^) and PMN-MDSC (CD11b^+^Ly-6G^+^Ly-6C^low^) recruited in the 4T1 model of breast cancer are T-cell-suppressive, beginning at d 28 post tumor injection [59]. This suggests that MDSC may have some other functional significance to the tumor before acquiring direct suppressive capacity, which may be recruitment of pro-tumor type 3 immune cells.

**Polarization to type 1 immunity.** We assessed recruitment of T_h_1/17 as a potential indicator of polarization from type 3 to the more beneficial type 1 immunity. The T_h_17 phenotype has been shown to be more plastic than the T_h_1 phenotype, and it is thought that T_h_1/17 represent cells transiting away from T_h_17 toward the T_h_1 phenotype [60]. Our data support this previous work, as we saw a robust and significant positive correlation between the presence of T_h_17 and T_h_1/17 in the tumor (Figure 4P). The lack of a similar correlation in peripheral tissues suggests, paradoxically, that the tumor microenvironment itself may be uniquely re-polarizing the immune response. We also noted negative correlations between T_h_1 and T_h_1/17 in peripheral tissues (Figure 4Q–T), further suggesting movement from type 3 to a type 1 immunity as T_h_1/17 polarize to replenish exhausted T_h_1 populations. Additionally, there was a rise in tumor T_h_1 at later time points (Figure 3H), which could reflect T_h_1/17 transiting toward T_h_1. It should be explored further whether the tumor specific milieu of cells and cytokines could be manipulated to encourage a unidirectional T_h_17 to T_h_1/17 to T_h_1 conversion, as this could be therapeutically beneficial. Nevertheless, it is difficult to rule out polarization from type 1 to type 3 immunity, as this would logically benefit the tumor. Consistent with this directionality are the negative correlations observed between both T_h_1 and T_h_17 and T_h_1 and T_h_22 in the spleen and tumor. We did not specifically examine which population of T_h_ cells (T_h_1 or T_h_17) was actively undergoing clonal expansion, but, again, based on previous work [60], we speculate that T_h_1/17 likely originate from T_h_17 and not T_h_1.

**Tumor-cell-specific IL-6 knockout.** We were interested in determining the mechanism underlying the changes in type 3 T_h_ and MDSC in the 4T1-WT time course studies described above. We focused on tumor-derived IL-6 expression for multiple reasons: (1) it can act as an immunosuppressive, pro-tumor cytokine expressed by 4T1 cells [61,62]; (2) it is a known driver of MDSC expansion [32]; (3) it has been implicated in type 3 immunity in other systems [51]; and (4) there exist IL-6-specific, FDA-approved monoclonal antibodies, tocilizumab and siltuximab, to treat rheumatoid arthritis [63] and Castleman disease [64], respectively. By using CRISPR to knock out IL-6 in the 4T1 cell line (4T1-IL6-KO), a unique opportunity was provided to evaluate the effect of excess IL-6 originating specifically from the tumor cells. Again, samples obtained from whole tumor tissue show that abundant stromal IL-6 expression accompanies the 4T1-IL6-KO cells (Figure 7C).

When assessing tumor-resident T_h_17 and T_h_22 in the 4T1-IL6-KO, T_h_17 was expanded, whereas T_h_22 was significantly reduced, consistent with other reports that IL-6 is a key cytokine determinate of both type 3 lineages [33]. However, IL-6 appears to be necessary but not sufficient to discriminate between these lineages, as the residual populations of these cells points to additional regulatory factors. One such factor is TGF-β1, which is thought to be a primary determinant of T_h_17 or T_h_22 polarization [33]. We did not assess whether the IL-6 KO altered TGF-β1 expression, but our data do suggest that the source and concentration of IL-6 influences T_h_22 or T_h_17 skewing. This is a mechanism that should be examined further, as there is little information on IL-6 source, TGF-β1 signaling, and IL-6 receptor expression in these type 3 cells or the other local constituents with which they would interact. 

While growth of the tumors was not impacted, 4T1-IL6-KO was associated with several immune effects that could be beneficial, if translatable to the oncology clinic. First, we saw a reduction in tumor resident T_h_22, a primary source of IL-22 that has been implicated in promoting growth and metastasis of various cancer types [65]. Reduced IL-6 from tumor cells also led to increased tumor-infiltrating T_h_17 and IFN-γ positive T_h_1/17, supporting the notion that these cells may be transitioning from T_h_17 to a T_h_1 phenotype; however, it is important to note that these cells may also give rise to T_reg_ [60,66,67], which were not assessed in the present study. Furthermore, we observed a reduction in M-MDSC across all tissues, including tumor tissues, and a significant reduction in PMN-MDSC in all peripheral organs. Indeed, this was our initial prediction and impetus for the IL-6 portion of the study. Other studies have shown that depletion of MDSC with guadecitabine encourages anti-tumor T-cell-mediated immunity, suggesting that the removal of MDSC alone is enough to bolster T cell-mediated immunotherapy [68,69]. Finally, and perhaps most unexpectedly and remarkably, was that 4T1-IL6-KO was associated with an increased total of CD4^+^ T_h_ cells in the tumors and blood. We are unaware of any other reports indicating that tumor-derived IL-6 could have such a broad, suppressive (or distractive) effect on T_h_ populations. Again, this and the other preclinical observations discussed above strongly support further exploration of IL-6-lowering monoclonal antibodies in the clinical oncology setting, with particular consideration given to combining these agents with checkpoint inhibitors and other type 1-dependent immunotherapies. 

**Limitations.** We acknowledge that this study has potential limitations, some of which we highlight here. Foremost, we did not follow up our study with a complete IL-6 knockout mouse. This means that other infiltrating stromal and immune cells were a source of IL-6; our goal, herein, was strictly to eliminate cancer-cell-derived IL-6. Similarly, we did not examine serum levels of IL-6 or other cytokines during development of these tumors, since targeting systemic IL-6 was not the objective of this study. Nevertheless, this could provide further insight on the effects tumoral IL-6 has in this model. Based on our data, we suggest that there are meaningful systemic changes in the cellular immune response to the removal of tumoral IL-6. Further, the nature of direct interactions, or the lack thereof, between the MDSC and type 3 T_h_ subsets will require manipulation studies directed at this question. While the trends between MDSC and T_h_17 and T_h_22 in the tumor are not significant, both co-exist in time and location, raising the possibility of reciprocal influence. Finally, we limited the scope of our study to the tracking of specific T_h_ subsets and MDSC, therefore, we do not know if any other changes to other immune cells occurred; given the broad influence of the T_h_ compartment over homeostasis, it is reasonable to hypothesize that other cellular parameters were affected. Additionally, given this was an observational study that did not investigate the functional consequences of type 3 immunity in tumors, it remains unclear as to whether the subsets play a consequential role in tumor growth by interacting with MDSC in peripheral organs. However, this work serves—to our knowledge—as one of the first to highlight the presence of type 3 immune cells in the 4T1 system. This should be considered when evaluating the immune system wholistically, whilst designing immunotherapeutic strategies in the presence of malignant disease.

In sum, this study demonstrated that there is a dynamic pro-tumor type 3 immune response in reaction to 4T1 mammary carcinoma. Furthermore, the effects of tumor-derived IL-6 on the tumor and peripheral immune landscapes were explored. It was found that knocking out tumor IL-6 significantly increased tumor-infiltrating T_h_ cells, significantly increased IFN-γ producing T_h_1/17, and reduced MDSC recruitment. These results demonstrate that IL-6 should be considered a therapeutic target in select circumstances where aberrant type 3 immune skewing is involved, which may better facilitate an initial, appropriate anti-tumor immune response. 

## 4. Methods and Materials

### 4.1. Mice and 4T1 Mammary Tumors

The 4T1 cells were purchased from ATCC (Manassas, VA, USA) and cultured in RPMI 1640 (ThermoFisher, Waltham, MA, USA) supplemented with 1% pen/strep, 2 mM L-Glutamine, 1 mM Sodium Pyruvate, 10 mM HEPES, 0.05 mM β-mercaptoethanol, and 10% (by volume) FB Essence (VWR, Radnor, PA, USA). To grow syngeneic, orthotopic mammary tumors, female BALB/cJ mice (Jackson Laboratory, Bar Harbor, ME, USA) between four to six weeks of age were inoculated with 1.0 × 10^4^ 4T1 cells. Cells were washed and suspended in 100 µL 1X HBSS (Ca^2+^/Mg^2+^ free, ThermoFisher) before injection through the nipple of the upper-right mammary fat pad using an insulin syringe. Mice were monitored daily for availability of food and water and any signs or symptoms of peripheral infection or inflammation. Palpable tumors were routinely detected 2 weeks after injection, and tumor sizes were monitored by calipers. Final tumor volumes were calculated by volume = (3.14/6)width × length^2^ [70]. In the 4T1 tumor time-course study, tissues were harvested at days 0, 7, 14, 21, and 28 post-injection (Figure 2A). On the day of tissue harvest, mice were euthanized by CO_2_ asphyxiation and tumors weighed before further processing. All animal procedures were performed according to approved Institutional Animal Care and Use Committee protocols (numbers: 1906CE-RH-RM-22 and 1702C-NP-M-20).

### 4.2. il6 Gene Knock-Out in 4T1 Cells

The CRISPR/Cas9 system was used to generate 4T1 cells deficient in IL-6 (4T1-IL6-KO). Guide RNAs (gRNA) were designed with CRISPOR software (http://crispor.tefor.net/, version 5.01 as of publication date, by Jean-Paul Concordet and Maximilian Haeussler, Santa Cruz, CA, USA, accessed on 15 April 2019) to target exon 2 of the *Mus musculus il6* gene (GenBank: M24221.1) at sequence TATACCACTTCACAAGTCGG. The gRNA oligos (Invitrogen, Carlsbad, CA, USA) were annealed, phosphorylated, and cloned into the pX458 vector following the Zhang lab protocol [71]. The vector pSpCas9(BB)-2A-GFP (PX458) was a gift from Feng Zhang (Addgene plasmid number 48138, Watertown, MA, USA). Plasmids were transfected using Lipofectamine 3000 (ThermoFisher). The vector includes a green fluorescent protein (GFP) reporter, and GFP^+^ cells were single-cell sorted into 96-well plates using a Sony SH800 Cell Sorter (Sony Biotechnology, San Jose, CA, USA). Subclones were cultured, and IL-6 deficiency was verified by fixed-cell intracellular flow cytometry with IL-6 antibody (BioLegend, #504504, San Diego, CA, USA). Two independent subclones verified subsequently as GFP-negative, IL-6 knockouts were pooled to create the 4T1-IL6-KO line used in these studies.

### 4.3. Tissue Processing

Spleens were weighed, dissected, and dissociated in 500 µL of 1X HBSS. Blood was collected from the chest cavity, following a cut to the aorta and local heparin infusion. Blood was centrifuged at 200× *g* for 10 min, and buffy coat and plasma were transferred to fresh vials, before lysing residual red blood cells with ACK lysis buffer (Quality Biological, Gaithersburg, MD, USA). Bone marrow was flushed from excised femurs and tibias using HBSS. Final tumor sizes were measured prior to excision, and tumors were weighed after excision and removal of extraneous tissue. Tumor tissue was minced and mechanically separated using a cell dissociation sieve fitted with a 100 μm mesh screen and resuspended in 1 mL HBSS. Cells were resuspended in type IV collagenase (2 mg/1 mL) and DNase (0.1 mg/mL; Worthington Biochemical, Lakewood, NJ, USA) and then incubated at 37 °C rocking at 225 rpm for 1 h. Remaining erythrocytes from various tissues were cleared with ACK lysis buffer.

### 4.4. Flow Cytometry and Antibodies 

Cells were stained in flow cytometry buffer: 0.5% BSA in Ca^2+^/Mg^2+^ free 1X Dulbecco’s PBS (ThermoFisher). Flow reagents and antibodies were purchased from BioLegend (San Diego, CA, USA), unless stated otherwise, and catalog numbers of critical antibody reagents are included for reproducibility. Fc block (#101302) was used prior to staining with extracellular markers, in accordance with the protocol of the manufacturer. Any surface marker staining was performed prior to fixation with buffer (#420801), in accordance with the recommendation of the manufacturer. For intracellular markers, prior to cell surface staining and fixation, cells were treated with 1X brefeldin A (#420601) for 3–4 h at 37 °C to block intracellular protein trafficking. Cells were then stained for surface markers, fixed and permeabilized with buffer (#421002), and finally stained for intracellular targets, in accordance with the protocol of the manufacturer. Samples were analyzed using an Attune NxT cytometer (ThermoFisher); raw data were processed in FCS Express version 6 (De Novo Software, Pasadena, CA, USA). 

The biomarker profiles used to define cell types in these studies were as follows: M-MDSC (CD11b^+^Ly-6G^−^Ly-6C^hi^), PMN-MDSC (CD11b^+^Ly-6G^+^Ly-6C^low^), T_h_17 (CD3^+^CD4^+^RORγt^+^IFN-γ^−^IL-17A^+^IL-22^+/−^), T_h_1/17 (CD3^+^CD4^+^RORγt^+^IFN-γ^+^IL-17A^+^IL-22^+/−^), T_h_1 (CD3^+^CD4^+^IFN-γ^+^RORγt^−^IL-17A^−^IL-22^+/−^), and T_h_22 (CD3^+^CD4^+^IFN-γ^−^RORγt^−^IL-17A^−^IL-22^+^). Antibodies for these markers included CD3 (#100237), CD4 (#100438), MDSC antibody cocktail (#77496), IFNγ (#505806), IL-22 (#516409), IL-17A (#506904), and RORγt, which were from ThermoFisher (#46698180). An example of flow cytometric gating strategy is presented in Figure 2B,C. 

### 4.5. Statistical Analyses

All data are presented as mean ± SEM, and statistical tests used α = 0.05. Grubbs’ test was used to identify outliers. All multiple comparisons were calculated using one-way ANOVA with Tukey’s multiple comparisons test. Single comparisons were calculated using Student’s t-test. A simple linear regression and Pearson’s correlation were used to assess the relationships between cell types in the 4T1-WT time course study; for each of these regressions, animals across all time points were included. A confidence interval (CI) of 95% is represented as well as Pearson r and r^2^ values, with significance set by α = 0.05. All statistical analyses were conducted in Prism 8 (GraphPad, San Diego, CA, USA). R^2^ (coefficient of determination) and r (Pearson correlation coefficient) values are used to predict the relationship between independent and dependent variables in a linear model. R represents the overall strength and direction of a correlation, while r^2^ is indicative of how well the data fit to a regression line. R values are measured from −1 to +1; 0 indicates no relationship. R^2^ is measured from 0 to +1, while 0 indicates the model explains none of the variability [72]. 

## Figures and Tables

**Figure 1 ijms-23-10299-f001:**
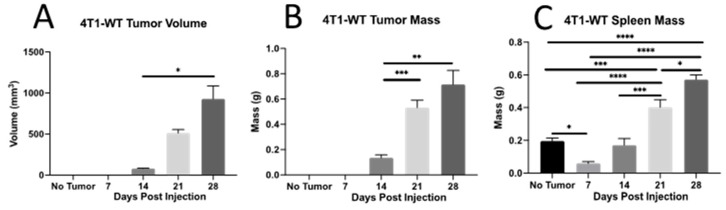
Tumor and spleen sizes in 4T1 tumor-bearing animals. The 4T1-WT mammary carcinoma cells (10^4^) were engrafted into the mammary gland of BALB/cJ mice, and animals were assessed for (**A**) tumor volume, (**B**) tumor mass, and (**C**) spleen mass at the times indicated. Data are represented as mean ± SEM; n = 4–11 mice per time point. * *p* < 0.05, ** *p* < 0.01, *** *p* < 0.001, and **** *p* < 0.0001.

**Figure 2 ijms-23-10299-f002:**
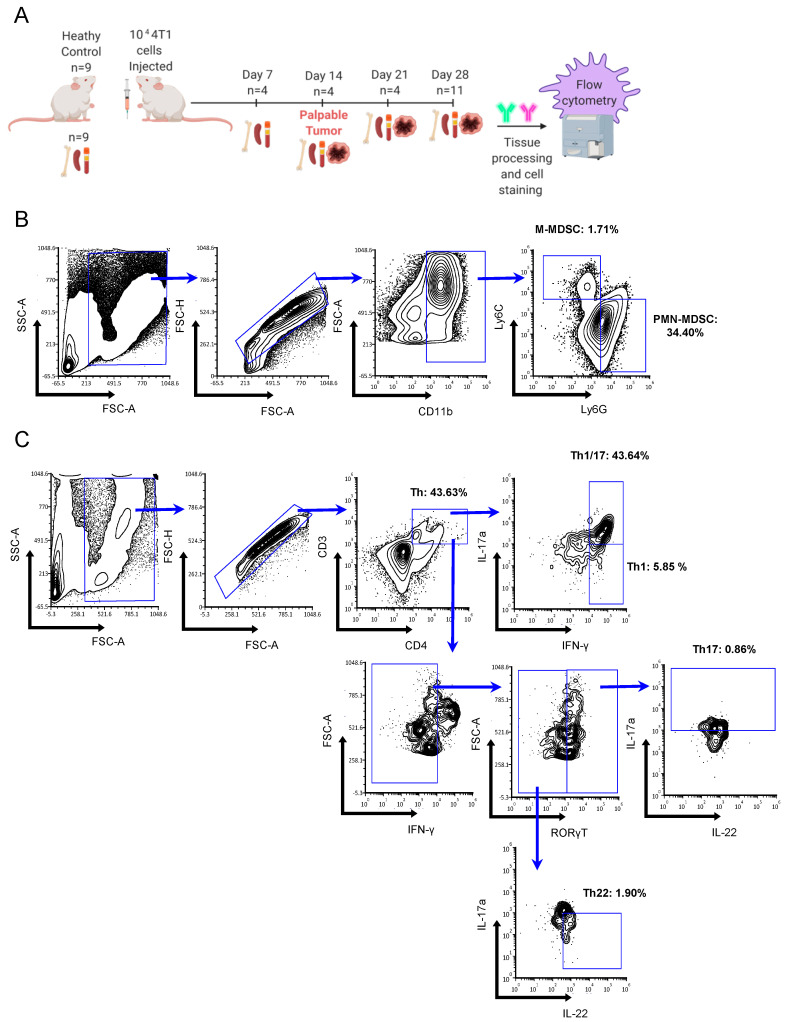
Time-course methodology and flow cytometry gating strategy. (**A**) Wild type 4T1 (4T1-WT, 10^4^ cells) were injected into syngeneic BALB/cJ mice and spleen, blood, bone marrow, and tumor (when present) and were harvested 7, 14, 21, and 28 days later. The 4T1 mice formed palpable tumors 14 d after injection. Tissue was processed, stained, and analyzed by flow cytometry, with age-matched healthy non-tumor-bearing mice included for comparison. Representative blood sample shows gating strategies for (**B**) MDSC subtypes M-MDSC (CD11b^+^Ly-6G^−^Ly-6C^hi^) and PMN-MDSC (CD11b^+^Ly-6G^+^Ly-6C^low^) as well as (**C**) CD3^+^CD4^+^ T helper (T_h_) subtypes T_h_1 (CD3^+^CD4^+^IFN-γ^+^RORγt^−^IL-17A^−^IL-22^+/−^), T_h_1/17 (CD3^+^CD4^+^RORγt^+^IFN-γ^+^IL-17A^+^IL-22^+/−^), T_h_17 (CD3^+^CD4^+^RORγt^+^IFN-γ^−^IL-17A^+^IL-22^+/−^), and T_h_22 (CD3^+^CD4^+^IFN-γ^−^RORγt^−^IL-17A^−^IL-22^+^).

**Figure 3 ijms-23-10299-f003:**
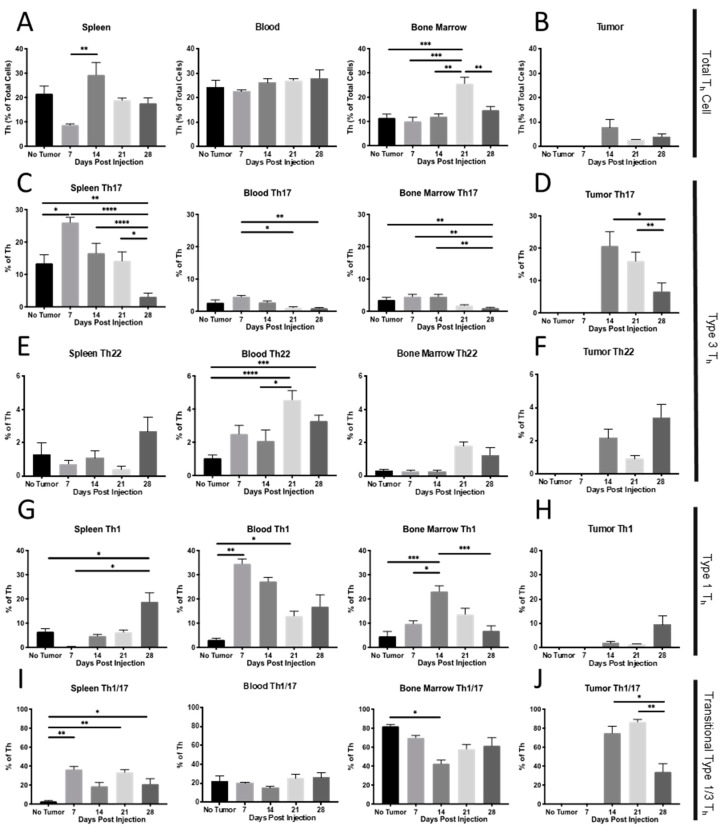
T_h_ populations over time in 4T1-WT mammary tumor bearing mice. Flow cytometric measurement of total T_h_, T_h_17, T_h_22, T_h_1, and T_h_1/17 in peripheral tissues (**A**,**C**,**E**,**G**,**I**) or tumors (**B**,**D**,**F**,**H**,**J**) at the indicated times. Data are presented as mean ± SEM; n = 4–11 mice per time point. * *p* < 0.05, ** *p* < 0.01, *** *p* < 0.001, and **** *p* < 0.0001.

**Figure 4 ijms-23-10299-f004:**
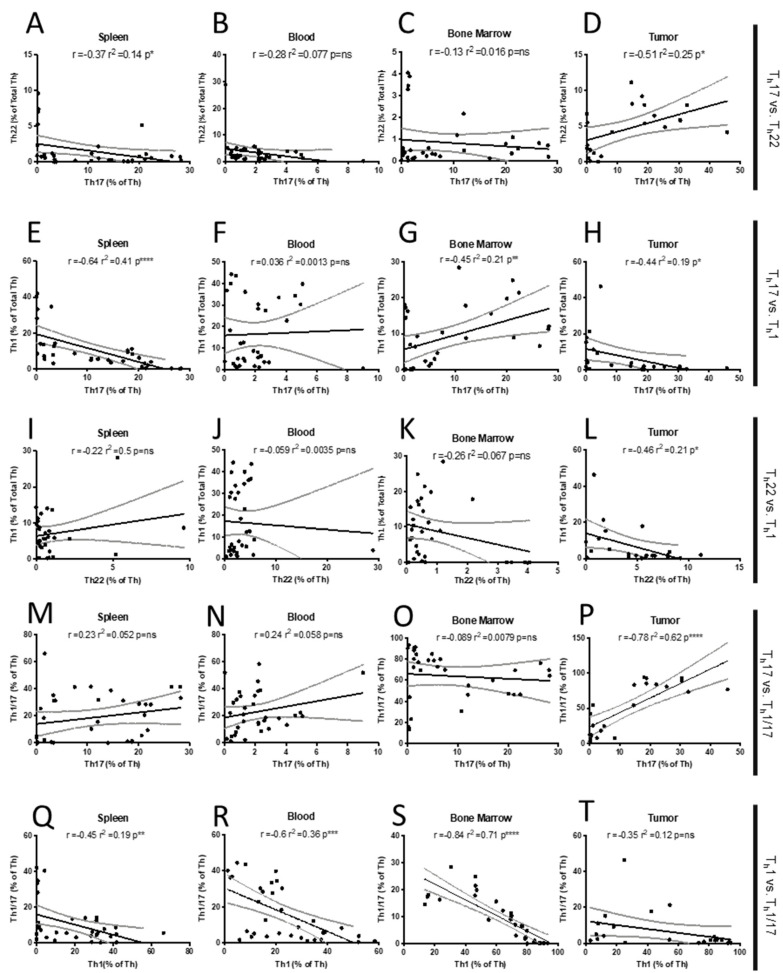
Correlations among T_h_ subsets in 4T1-WT tumor-bearing mice. Pearson correlations between (**A**–**D**) T_h_17 and T_h_22, (**E**–**H**) T_h_17 and T_h_1, (**I**–**L**) T_h_22 and T_h_1, (**M**–**P**) T_h_17 and T_h_1/17, (**Q**–**T**) and T_h_1 and T_h_1/17 levels in spleen, blood, bone marrow, and tumors in 4T1-WT tumor-bearing mice. Simple linear regression and 95% CI are included to help visualize relationships. *p* = ns (not significant), * *p* < 0.05, ** *p* < 0.01, *** *p* < 0.001, and **** *p* < 0.00001.

**Figure 5 ijms-23-10299-f005:**
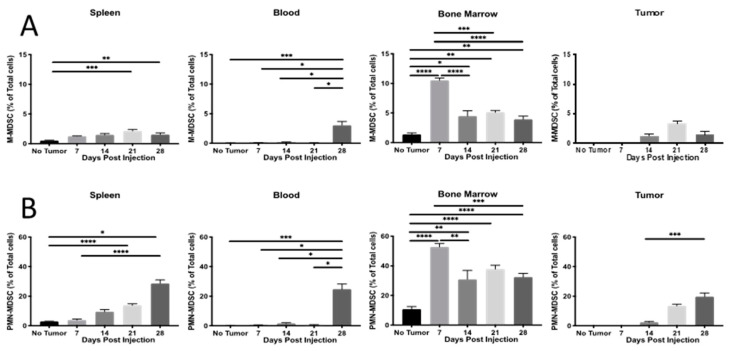
4T1-WT mammary-tumor-induced MDSC expansion. Flow cytometric measurement of (**A**) M-MDSC and (**B**) PMN-MDSC in response to 4T1-WT mammary tumor development over time. Data are presented as mean ± SEM and n = 4–11 mice per time point. * *p* < 0.05, ** *p* < 0.01, *** *p* < 0.001, and **** *p* < 0.0001.

**Figure 6 ijms-23-10299-f006:**
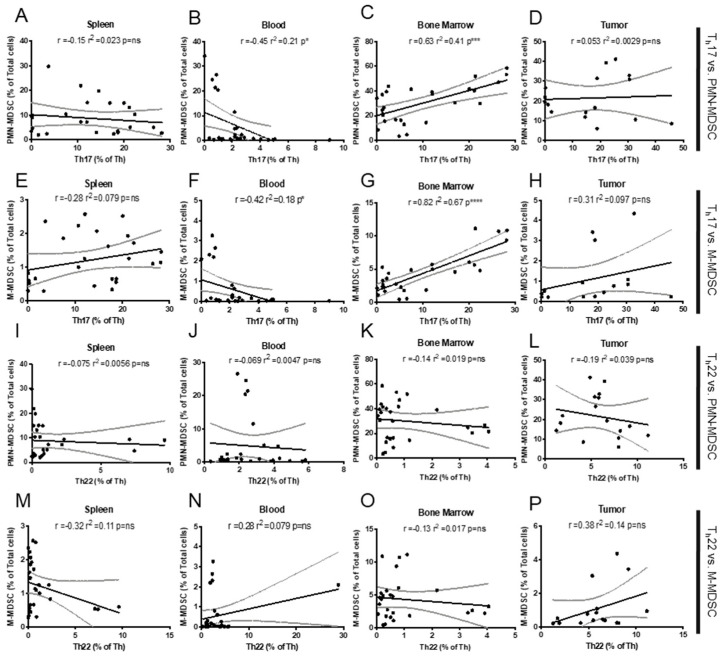
Correlations between type 3 T_h_ and MDSC accumulation. Pearson correlation assessing the association between type 3 T_h_ percentages is shown in Figure 3, and MDSC percentages are shown in Figure 5. (**A**–**D**) T_h_17 and PMN-MDSC, (**E**–**H**) T_h_17 and M-MDSC, (**I**–**L**) T_h_22 and PMN-MDSC, and (**M**–**P**) T_h_22 and M-MDSC recruitment in the spleen, blood, bone marrow, and tumor of 4T1 tumor-bearing animals. Simple linear regression included to visualize relationship and 95% CI are presented. *p* = ns (not significant), * *p* < 0.05, *** *p* < 0.001, and **** *p* < 0.00001.

**Figure 7 ijms-23-10299-f007:**
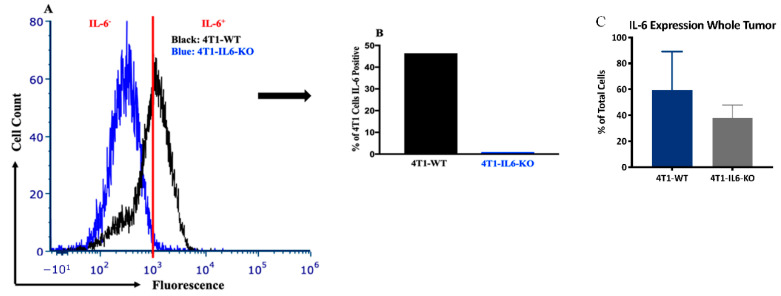
CRISPR/Cas9 knockout of IL-6 in the 4T1 cell line. Wild type 4T1 (4T1-WT) vs. IL-6 knock out 4T1 (4T1-IL6-KO) expression of IL-6 via intracellular flow cytometry. (**A**) Histogram demonstrating relative fluorescence, and (**B**) percentage of total 4T1 that are IL-6 positive. (**C**) Total cells positive for IL-6 in tumors grown from 4T1-WT and 4T1-IL6-KO lines. Background IL-6 positivity in KO tumors likely derives from tumor-associated stromal cells.

**Figure 8 ijms-23-10299-f008:**
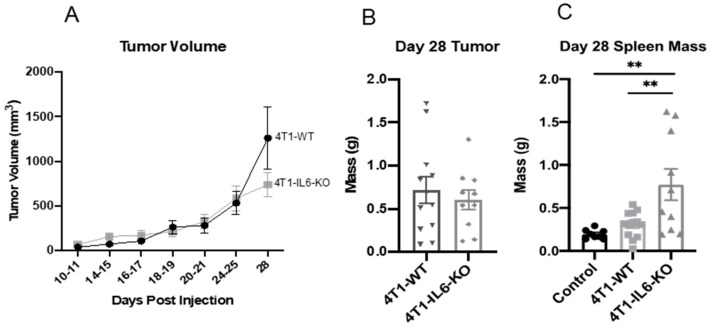
Tumor characteristics and spleen mass in 4T1–WT vs. 4T1–IL6–KO mice. (**A**) The 4T1-WT and 4T1-IL6-KO tumor volumes were measured over time; no statistically significant differences (*p >* 0.05) were detected. (**B**) Final 4T1-WT and 4T1-IL6-KO tumor mass at d 28 post engraftment. (**C**) Spleen mass at d 28 following tumor engraftment; healthy non-tumor-bearing spleens included for reference. The 4T1-IL6-KO spleens were significantly heavier than 4T1-WT and non-tumor-bearing controls. Note that 4T1-IL6-KO is a 4T1-cell-specific IL-6 knockout, not whole animal. ANOVA with Tukey’s multiple comparisons. Data are presented as mean ± SEM and n = 6–11 mice/group. ** *p* < 0.01.

**Figure 9 ijms-23-10299-f009:**
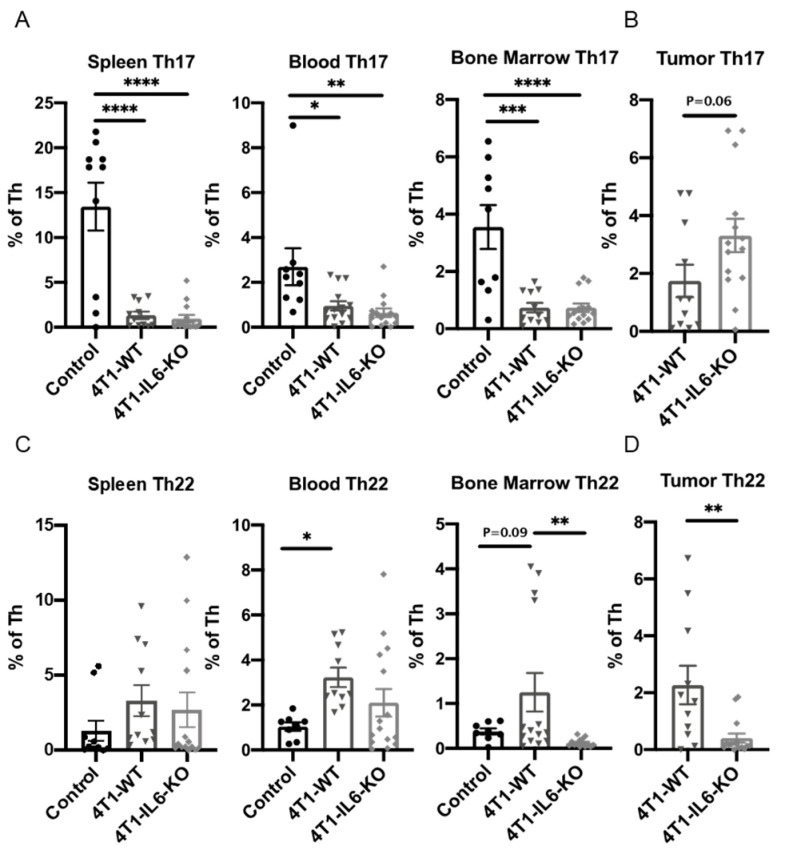
Knocking out tumor cell IL-6 production significantly alters type 3 T helper cell recruitment in the tumor and peripheral tissues. Tumors and peripheral tissues harvested 28 d after engraftment of 4T1-WT and 4T1-IL6-KO cell lines were analyzed for proportions of (**A**,**B**) T_h_17 and (**C**,**D**) T_h_22 type 3 T_h_ cells. Healthy, non-tumor bearing mice were analyzed for comparison (Control); see Figure 2 for gating strategy. Statistical significance was measured using one-way ANOVA with Tukey’s multiple comparisons or Student’s t-test. Data are presented as mean ± SEM with n = 4–11 mice/group. * *p* < 0.05, ** *p* < 0.01, *** *p* < 0.001 and **** *p* < 0.0001.

**Figure 10 ijms-23-10299-f010:**
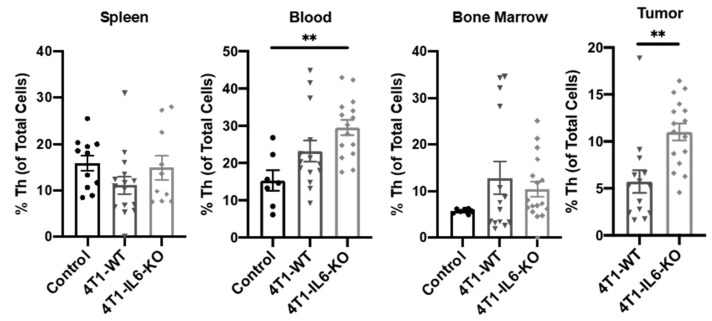
Knocking out tumor cell IL-6 production increases tumor-infiltrating CD3^+^/CD4^+^ T helper cells. Tumors and peripheral tissues harvested 28 d after engraftment of 4T1-WT and 4T1-IL6-KO cell lines were analyzed for total CD3^+^CD4^+^ T_h_ cells. Healthy, non-tumor-bearing mice were analyzed for comparison (control); see Figure 2 for gating strategy. Data are presented as mean ± SEM; n = 4–11 mice/group. ** *p* < 0.01.

**Figure 11 ijms-23-10299-f011:**
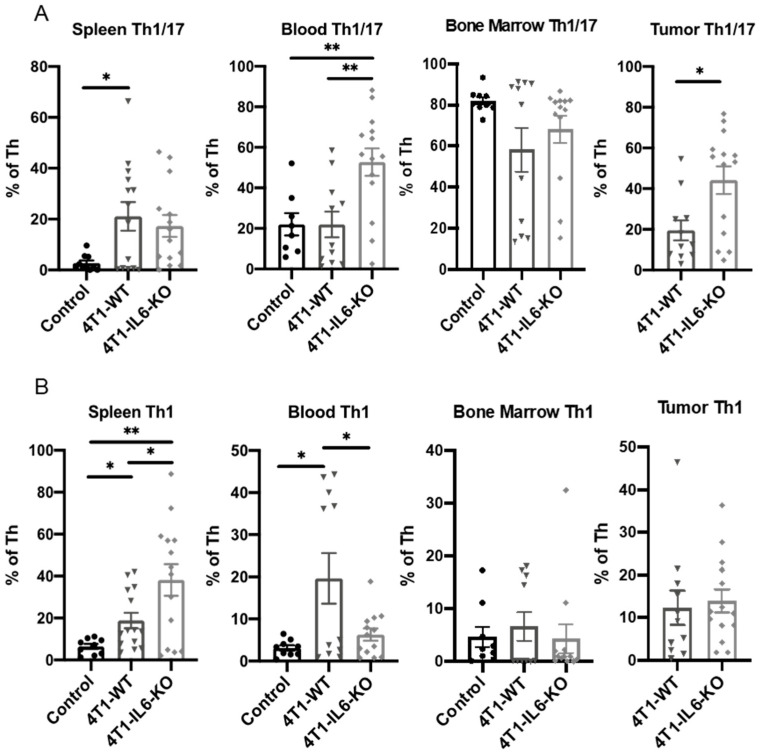
Depleting tumor cell IL-6 increases tumor-infiltrating T_h_1/17 cells. Tumors and peripheral tissues harvested 28 d after engraftment of 4T1-WT and 4T1-IL6-KO cell lines were analyzed for proportions of (**A**) T_h_1/17 and (**B**) T_h_1 subtype T helper cells. Healthy, non-tumor-bearing mice were analyzed for comparison (control); see Figure 2 for gating strategy. Data are presented as mean ± SEM; n = 4–11 mice/group. * *p* < 0.05, ** *p* < 0.01.

**Figure 12 ijms-23-10299-f012:**
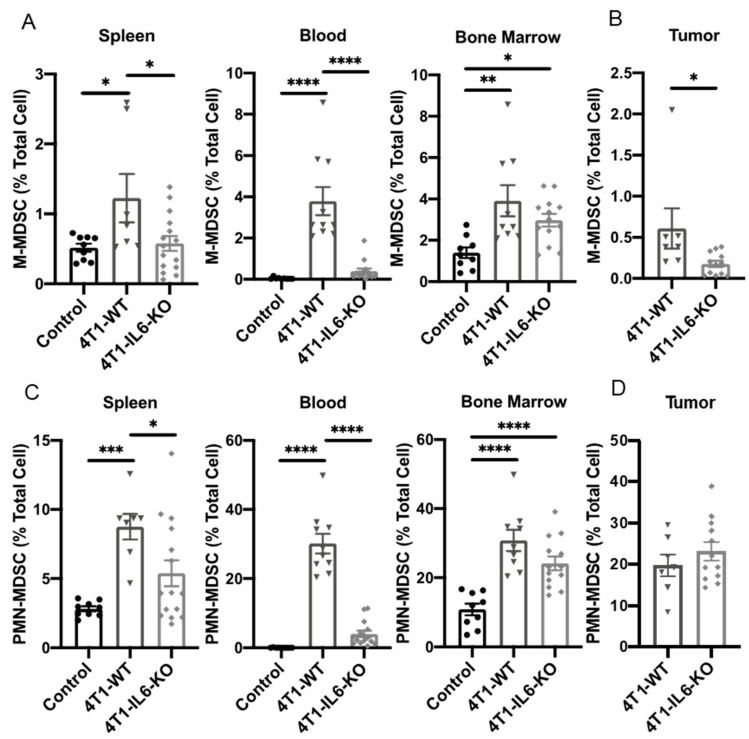
Knocking out tumor cell IL-6 production significantly reduces M-MDSC and PMN-MDSC recruitment. Tumors and peripheral tissues harvested 28 d after engraftment of 4T1-WT and 4T1-IL6-KO cell lines were analyzed for proportions of (**A**,**B**) M-MDSC and (**C**,**D**) PMN-MDSC. Healthy, non-tumor-bearing mice were analyzed for comparison (control); see Figure 2 for gating strategy. Data are presented as mean ± SEM; n = 4–11 mice/group. * *p* < 0.05, ** *p* < 0.01, *** *p* < 0.001, and **** *p* < 0.0001.

## Data Availability

Analyzed data are included in this manuscript. Raw data are available from the authors upon request.

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
