# Peer review of "Th17, Th22, and Myeloid-Derived Suppressor Cell Population Dynamics and Response to IL-6 in 4T1 Mammary Carcinoma"

_ijms, 2022, doi:10.3390/ijms231810299_

Round 1

Reviewer 1 Report

Comments:

The authors attempt in this study to establish a correlation between type 3 immunity, specifically Th17 and Th22 cells, and myeloid-derived suppressor cells (MDSC) and related resistance to immunotherapy. Using the well-established 4T1 mouse mammary tumor, the investigators assess growth of orthotopic tumors and monitor relative proportions of Th1, Th17, Th22, and both monocytic (M)-MDSC and granulocytic (PMN)-MDSC in bone marrow, blood, spleen, and tumor at defined time points through multi-parameter flow cytometry. They compare wild type 4T1 with 4T1 in which IL-6 is genetically deleted. Although 4T1 and 4T1-IL-6-KO tumors grew similarly in mice, the authors conclude that IL-6 drives expansion of Th17 and Th22 that may facilitate tumor growth and confer immunotherapy resistance. Overall, this study offers limited and mostly descriptive data that make drawing solid conclusions challenging. Specific comments include:

1. The observed trends in relative proportion of Th subsets over time are somewhat confusing. For example, Th17 as a percentage of Th decline in tumors over time whereas Th22 tends to increase over time, but the d.21 values for Th22 are substantially lower than d.14 and d.28. It is not clear why.

2. The authors state on p.9 that their data are consistent with Th differentiation away from Th1 towards Th17; however, this trend was not significant in tumors. One could argue that if this was indeed a driver of tumor progression, this trend would be most significant in tumors.

3. Earlier studies have demonstrated the ability of MDSC, in particular PMN-MDSC to drive progression of 4T1 tumors. In the current study the authors find no correlation between Th22 and M- or PMN-MDSC in tumors (or any other compartment). Likewise, there is no correlation between Th17 and MDSC in the tumor (Fig 6), which questions the role of type 3 immunity in 4T1 tumor progression.

4. The authors state on p.11 that their time-course data strongly suggest MDSC and type 3 Th cells coexist at elevated proportions. This is misleading as Fig 6 shows no correlation between MDSC and Th22 in any compartment, and Th17 only correlates with MDSC in blood and bone marrow, but the correlation in blood is a negative correlation.

5. The authors attempt to study the role of IL-6 in driving changes in proportions of MDSC and type 3 Th cells by using 4T1 and 4T1-IL-6-KO tumor cells, but in vivo IL-6 knock out tumors express a similar amount of IL-6 as wild type tumors. This seems to defeat the purpose of the proposed studies.

6. Data from fig 12 suggest that IL-6 may promote M-MDSC, but since the proportion of M-MDSC is very small (~0.5% of total cells) compared to PMN-MDSC (~20% of total cells), the biological impact is questionable.

7. It is difficult to understand the biological impact of IL-6 and type 3 immune subsets without analysis of other immune cell subsets, such as CD8 T cells, TAM, dendritic cell subsets, and Treg.

Author Response

The authors attempt in this study to establish a correlation between type 3 immunity, specifically Th17 and Th22 cells, and myeloid-derived suppressor cells (MDSC) and related resistance to immunotherapy. Using the well-established 4T1 mouse mammary tumor, the investigators assess growth of orthotopic tumors and monitor relative proportions of Th1, Th17, Th22, and both monocytic (M)-MDSC and granulocytic (PMN)-MDSC in bone marrow, blood, spleen, and tumor at defined time points through multi-parameter flow cytometry. They compare wild type 4T1 with 4T1 in which IL-6 is genetically deleted. Although 4T1 and 4T1-IL-6-KO tumors grew similarly in mice, the authors conclude that IL-6 drives expansion of Th17 and Th22 that may facilitate tumor growth and confer immunotherapy resistance. Overall, this study offers limited and mostly descriptive data that make drawing solid conclusions challenging. Specific comments include:

We thank the reviewer for their time spent reviewing and critiquing our work. The criticisms offered are apt and we do not disagree with any of the points. While the results presented are mostly descriptive, we believe the relative dearth of documented knowledge in this area warrants the work, specifically when considering the tailoring of novel immunotherapies to specific cancer or patient scenarios. We propose that these changes in immune cell compartments could have substantive influence over the efficiency of some therapeutic approaches. Regretfully, we are unable to address the concerns of the reviewer at scale (given response time and funding), thus we have added a new limitations section as the penultimate paragraph (lines 527-548, p. 18), where we specifically acknowledge the shortcomings of the work. These are in turn good opportunities for future, sustained research by us and other independent investigators.

  1. The observed trends in relative proportion of Th subsets over time are somewhat confusing. For example, Th17 as a percentage of Th decline in tumors over time whereas Th22 tends to increase over time, but the d.21 values for Th22 are substantially lower than d.14 and d.28. It is not clear why.

In designing the initial studies presented here, our aim was to characterize relative changes in Type 3 immune cell populations in the hopes of eventually identifying the mechanism. We agree with the reviewer that understanding the mechanisms behind the observed changes in Type 3 Th cells would be interesting; however, we do not have information on the mechanisms beyond the changes observed with the IL-6 knockout 4T1 cells employed for the experiments shown in the later figures.

  1. The authors state on p.9 that their data are consistent with Th differentiation away from Th1 towards Th17; however, this trend was not significant in tumors. One could argue that if this was indeed a driver of tumor progression, this trend would be most significant in tumors.

We do not disagree with the reviewer. However, there are substantive changes in systemic immune compartments, which we think could reasonably affect either tumor response and/or overall health of the host. We have acknowledged this in the new limitations section.

  1. Earlier studies have demonstrated the ability of MDSC, in particular PMN-MDSC to drive progression of 4T1 tumors. In the current study the authors find no correlation between Th22 and M- or PMN-MDSC in tumors (or any other compartment). Likewise, there is no correlation between Th17 and MDSC in the tumor (Fig 6), which questions the role of type 3 immunity in 4T1 tumor progression.

We acknowledge the lack of correlation as a limitation, however it is clear that both Th and MDSC are present and elevated, despite lack of statistical correlation. The need for additional studies to clarify the functional consequence, if any, of these mutually elevated immune populations is addressed in the limitations section.  

  1. The authors state on p.11 that their time-course data strongly suggest MDSC and type 3 Th cells coexist at elevated proportions. This is misleading as Fig 6 shows no correlation between MDSC and Th22 in any compartment, and Th17 only correlates with MDSC in blood and bone marrow, but the correlation in blood is a negative correlation.

We have addressed this comment by changing the language in text at lines 302-306 (p. 11) as follows:

“Our time-course data suggested that MDSC and type 3 Th cells exist at elevated proportions in systemic immunological tissues of tumor-bearing animals, specifically both MDSC and Th17 were elevated in blood and bone marrow.  Furthermore, there was some indication from the bone marrow that MDSC presence was influencing Th17 polarization.”

  1. The authors attempt to study the role of IL-6 in driving changes in proportions of MDSC and type 3 Th cells by using 4T1 and 4T1-IL-6-KO tumor cells, but in vivoIL-6 knock out tumors express a similar amount of IL-6 as wild type tumors. This seems to defeat the purpose of the proposed studies.

We initially hypothesized that excess IL-6, specifically sourced from the tumor, was solely responsible for the marked changes in Type 3 Th cells observed in our initial experiments. We thus focused on tumor-derived IL-6 . While IL-6 is a starting point, we agree with the reviewer, and even expected, that other sources of IL-6 are present. We believe this is likely due to localized stromal cells and other invading leukocytes such as mast cells. However, we are not fully confident in the additional sources of IL-6 at this time, and it is an active area of study.

  1. Data from fig 12 suggest that IL-6 may promote M-MDSC, but since the proportion of M-MDSC is very small (~0.5% of total cells) compared to PMN-MDSC (~20% of total cells), the biological impact is questionable.

We do not disagree with the reviewer on this point. However, these numbers are in line with other studies in the field, and it is reasonable for relatively rare events (as another example: Treg among others) to have substantive impact in a system.

  1. It is difficult to understand the biological impact of IL-6 and type 3 immune subsets without analysis of other immune cell subsets, such as CD8 T cells, TAM, dendritic cell subsets, and Treg.

We have acknowledged the lack of mechanistic detail as a limitation.

Reviewer 2 Report

This is an interesting paper that attempts to show an induction relationship between tumor-local Type 3 helper T cells and MDSCs. Furthermore, it is significant as basic research for clinical application to cancer treatment by neutralizing IL-6. However, in order to prove that IL-6 produced by tumor cells has an effect on Type 3 helper T cells and MDSCs, I think that additional results to be demonstrated as described below are necessary.

Major points

1) What kind of cytokines/chemokines were elevated in the blood after transplantation of 4T1 cells into mice, and what kind of cytokine/chemokines were changed when 4T1-IL-6-KO cells were transplanted? Authors should show these results. Without confirming that 4T1 cell transplantation increases IL-6 in the blood and that 4T1-IL-6-KO cell transplantation does not increase IL-6, it would be difficult to describe the author's conclusion.

2) If 4T1 cells constitutively produce IL-6, it is possible that IL-6 autocrine also induces the production of other cytokines and chemokines. In 4T1-IL-6-KO cells, it will be necessary to confirm what kind of changes in cytokines and chemokines that are constantly secreted from 4T1 cells by IL-6-KO. The fact that only IL-6 is reduced may be necessary supporting evidence to argue that IL-6 is a therapeutic target.

Minor point

3) Authors used GFP as an indicator in the operation to eliminate IL-6 expression in 4T1 cells. However, the expression of GFP should be transient and disappears after establishment of 4T1-IL-6-KO cells. Because, if the expression of GFP remains in the cells, it may become a foreign antigen and easily induce an immune response in mice. Did authors check the disappearance of GFP? Also, authors used 4T1-WT cells as a control, which were similarly manipulated and subcloned?

Author Response

This is an interesting paper that attempts to show an induction relationship between tumor-local Type 3 helper T cells and MDSCs. Furthermore, it is significant as basic research for clinical application to cancer treatment by neutralizing IL-6. However, in order to prove that IL-6 produced by tumor cells has an effect on Type 3 helper T cells and MDSCs, I think that additional results to be demonstrated as described below are necessary.

We thank the reviewer for their time reading and offering thoughtful criticisms of our work. We agree that these major points are important, and indeed are areas we have often considered. However, given limitations of scale (time and funding) we chose to focus on what we feel are the cornerstone, novel observations, namely the Type 3 Th compartment. We have written a new limitations section (lines 527-548, p. 18) where we specifically acknowledge the gaps in this work, which we also see as opportunities for future study by us and other interested laboratories independently.

Major points

1) What kind of cytokines/chemokines were elevated in the blood after transplantation of 4T1 cells into mice, and what kind of cytokine/chemokines were changed when 4T1-IL-6-KO cells were transplanted? Authors should show these results. Without confirming that 4T1 cell transplantation increases IL-6 in the blood and that 4T1-IL-6-KO cell transplantation does not increase IL-6, it would be difficult to describe the author's conclusion.

We appreciate the reviewer raising these concerns regarding circulating concentrations of cytokines or chemokines in the presence of the 4T1 tumors. Unfortunately, these concentrations were not assayed in the experiments described here and would require significant additional efforts to address this question (essentially repeating the entire experiment). Therefore, we feel it is beyond the scope of the current paper. We initially hypothesized that excess IL-6, specifically sourced from the tumor, was solely responsible for the marked changes in Type 3 Th cells observed in our initial experiments, and thus focused on tumor-derived IL-6 . While IL-6 is a starting point, we agree with the reviewer that changes in circulating concentrations of other cytokines are at play, and a broad panel measuring other circulating cytokines will be very informative in future studies.

We have acknowledged this in our new limitations section.

2) If 4T1 cells constitutively produce IL-6, it is possible that IL-6 autocrine also induces the production of other cytokines and chemokines. In 4T1-IL-6-KO cells, it will be necessary to confirm what kind of changes in cytokines and chemokines that are constantly secreted from 4T1 cells by IL-6-KO. The fact that only IL-6 is reduced may be necessary supporting evidence to argue that IL-6 is a therapeutic target.

We agree with the reviewer that the lack of IL-6 production in these 4T1 cells could alter production of other cytokines by autocrine mechanisms. We did not pursue this specific question as part of these studies; however, we hope to consider this question as part of our ongoing studies with these cells. 

Minor point

 3) Authors used GFP as an indicator in the operation to eliminate IL-6 expression in 4T1 cells. However, the expression of GFP should be transient and disappears after establishment of 4T1-IL-6-KO cells. Because, if the expression of GFP remains in the cells, it may become a foreign antigen and easily induce an immune response in mice. Did authors check the disappearance of GFP? Also, authors used 4T1-WT cells as a control, which were similarly manipulated and subcloned?

We appreciate the reviewer raising these concerns. We sorted GFP-positive cells 48 hours following transient transfection since we observed a significant reduction in GFP expression in the mixed populations beyond the 48-hour time point. The two founding subclones selected to become the 4T1-IL6-KO used for this work were verified weeks later by fluorescent microscopy and flow cytometry as GFP-negative, suggesting there was not a stable integration of the Cas9-GFP reporter plasmid construct resulting from the transient transfection procedure. We have updated the methods on page 5 in the ‘IL-6 Gene Knock-Out in 4T1 Cells’ section to reflect that these cells were indeed GFP-negative.

In preliminary experiments with the Cas-9 IL-6-KO plasmid construct, we isolated numerous clonal lines that were GFP-positive initially, but later determined to have maintained IL-6 expression. These failed subclones, along with the successful clones that became the IL-6-KO cell line used here, were found to be indistinguishable from wild type 4T1 cells regarding their proliferation rate, morphology, and chemotaxis towards conditioned medium. Based on these early experimental observations, we felt confident moving forward with the unmanipulated 4T1-WT population as a comparison.

Round 2

Reviewer 1 Report

The authors decided not to perform any additional experiments but chose instead to add a paragraph describing the limitations of the study. While somewhat useful, it does not improve the quality and significance of the manuscript. Most of the observations are descriptive and provide at best a start point for further study.

Author Response

As stated in our original response, we cannot repeat the entire set of experiments at this time for analysis of the requested parameters/observations. We are substantively limited in both time and funding. We agree that these are all valuable points, and we have acknowledged this in Limitations.

Since the content of the reviewer's request is not different from the first review, it has been requested that our initial, detailed response be included here. Thus, from the first-round review:

Reviewer 1:

The authors attempt in this study to establish a correlation between type 3 immunity, specifically Th17 and Th22 cells, and myeloid-derived suppressor cells (MDSC) and related resistance to immunotherapy. Using the well-established 4T1 mouse mammary tumor, the investigators assess growth of orthotopic tumors and monitor relative proportions of Th1, Th17, Th22, and both monocytic (M)-MDSC and granulocytic (PMN)-MDSC in bone marrow, blood, spleen, and tumor at defined time points through multi-parameter flow cytometry. They compare wild type 4T1 with 4T1 in which IL-6 is genetically deleted. Although 4T1 and 4T1-IL-6-KO tumors grew similarly in mice, the authors conclude that IL-6 drives expansion of Th17 and Th22 that may facilitate tumor growth and confer immunotherapy resistance. Overall, this study offers limited and mostly descriptive data that make drawing solid conclusions challenging. Specific comments include:

We thank the reviewer for their time spent reviewing and critiquing our work. The criticisms offered are apt and we do not disagree with any of the points. While the results presented are mostly descriptive, we believe the relative dearth of documented knowledge in this area warrants the work, specifically when considering the tailoring of novel immunotherapies to specific cancer or patient scenarios. We propose that these changes in immune cell compartments could have substantive influence over the efficiency of some therapeutic approaches. Regretfully, we are unable to address the concerns of the reviewer at scale (given response time and funding), thus we have added a new limitations section as the penultimate paragraph (lines 527-548, p. 18), where we specifically acknowledge the shortcomings of the work. These are in turn good opportunities for future, sustained research by us and other independent investigators.

  1. The observed trends in relative proportion of Th subsets over time are somewhat confusing. For example, Th17 as a percentage of Th decline in tumors over time whereas Th22 tends to increase over time, but the d.21 values for Th22 are substantially lower than d.14 and d.28. It is not clear why.

In designing the initial studies presented here, our aim was to characterize relative changes in Type 3 immune cell populations in the hopes of eventually identifying the mechanism. We agree with the reviewer that understanding the mechanisms behind the observed changes in Type 3 Th cells would be interesting; however, we do not have information on the mechanisms beyond the changes observed with the IL-6 knockout 4T1 cells employed for the experiments shown in the later figures.

  1. The authors state on p.9 that their data are consistent with Th differentiation away from Th1 towards Th17; however, this trend was not significant in tumors. One could argue that if this was indeed a driver of tumor progression, this trend would be most significant in tumors.

We do not disagree with the reviewer. However, there are substantive changes in systemic immune compartments, which we think could reasonably affect either tumor response and/or overall health of the host. We have acknowledged this in the new limitations section.

  1. Earlier studies have demonstrated the ability of MDSC, in particular PMN-MDSC to drive progression of 4T1 tumors. In the current study the authors find no correlation between Th22 and M- or PMN-MDSC in tumors (or any other compartment). Likewise, there is no correlation between Th17 and MDSC in the tumor (Fig 6), which questions the role of type 3 immunity in 4T1 tumor progression.

We acknowledge the lack of correlation as a limitation, however it is clear that both Th and MDSC are present and elevated, despite lack of statistical correlation. The need for additional studies to clarify the functional consequence, if any, of these mutually elevated immune populations is addressed in the limitations section.  

  1. The authors state on p.11 that their time-course data strongly suggest MDSC and type 3 Th cells coexist at elevated proportions. This is misleading as Fig 6 shows no correlation between MDSC and Th22 in any compartment, and Th17 only correlates with MDSC in blood and bone marrow, but the correlation in blood is a negative correlation.

We have addressed this comment by changing the language in text at lines 302-306 (p. 11) as follows:

“Our time-course data suggested that MDSC and type 3 Th cells exist at elevated proportions in systemic immunological tissues of tumor-bearing animals, specifically both MDSC and Th17 were elevated in blood and bone marrow.  Furthermore, there was some indication from the bone marrow that MDSC presence was influencing Th17 polarization.”

  1. The authors attempt to study the role of IL-6 in driving changes in proportions of MDSC and type 3 Th cells by using 4T1 and 4T1-IL-6-KO tumor cells, but in vivoIL-6 knock out tumors express a similar amount of IL-6 as wild type tumors. This seems to defeat the purpose of the proposed studies.

We initially hypothesized that excess IL-6, specifically sourced from the tumor, was solely responsible for the marked changes in Type 3 Th cells observed in our initial experiments. We thus focused on tumor-derived IL-6 . While IL-6 is a starting point, we agree with the reviewer, and even expected, that other sources of IL-6 are present. We believe this is likely due to localized stromal cells and other invading leukocytes such as mast cells. However, we are not fully confident in the additional sources of IL-6 at this time, and it is an active area of study.

  1. Data from fig 12 suggest that IL-6 may promote M-MDSC, but since the proportion of M-MDSC is very small (~0.5% of total cells) compared to PMN-MDSC (~20% of total cells), the biological impact is questionable.

We do not disagree with the reviewer on this point. However, these numbers are in line with other studies in the field, and it is reasonable for relatively rare events (as another example: Treg among others) to have substantive impact in a system.

  1. It is difficult to understand the biological impact of IL-6 and type 3 immune subsets without analysis of other immune cell subsets, such as CD8 T cells, TAM, dendritic cell subsets, and Treg.

We have acknowledged the lack of mechanistic detail as a limitation.

Reviewer 2 Report

I understood the original focus of the authors and the initial purpose of the study. I look forward to future developments in the authors' research.

Author Response

We thank the reviewer for their comments and time, and look forward to this work stimulating future research on the topic.